# Research on financing countermeasures of online game SMEs based on the identification of intangible assets information

**Gaige Tu**[1]*, **Hao Chen**[1], **Chunxiao Zhu**[2]

**1** School of Business Administration, Zhongnan University of Economics and Law, Wuhan, Hubei, China,
**2** School of Business Administration, Henan Vocational College of Quality Engineering, Pingdingshan, Henan, China

* 18317069352@163.com

## Abstract

As an important part of the cultural industry, small and medium-sized online game enterprises undertake the functions of technological innovation, employment absorption and cultural cultivation. However, the lack of credit ability restricts the financial development of such enterprises. To solve the financing problem of online small and medium-sized game enterprises, this paper firstly uses the information of intangible assets to identify their credit ability, and considers that the information of intangible assets is a problem worthy of attention in evaluating credit risk. Secondly, the intangible assets information disclosure index, the revenue sharing contract of credit synergy and the dynamic game mechanism are constructed to study the importance of the intangible assets index and the evolution of the dynamic game. Finally, the empirical study shows that the intangible assets of delisting and special treated online game small and medium-sized enterprises still have value, this type of enterprise and credit suppliers have the behavior of seeking advantages and avoiding disadvantages. Therefore, credit synergy should be constructed and government regulation should be implemented.

## Introduction

China vigorously supports the development of emerging cultural industries, making the online game industry, a representative of emerging cultural formats, gradually become one of the important cultural industry pillars that promote economic growth. As an important market participant in the online game industry, online game small and medium-sized enterprises (SMEs) are widely distributed in various segments including game development and design and product services. They are one of the most active parts of the software market for innovative activities. Such enterprises have the characteristics of light assets and knowledge-intensive, and correspondingly lack physical collateral. This type of enterprise has developed competitive products or services through technological innovation in order to meet the needs that the diversification of consumption in the mobile network market and consumer demand for boutique game products, thereby establishing a certain market advantage to improve its own credit

**Data Availability Statement:** All relevant data is within the paper.

**Funding:** The authors received no specific funding for this work.

**Competing interests:** The authors have declared that no competing interests exist.

capacity. However, online game SMEs usually encounter exogenous financing obstacles in the early stages of their development. Some banks and third-party financial institutions even include such companies on its negative list. Simultaneously, large online game companies squeeze their credit. The limited self-owned capital of an enterprise cannot guarantee its sustainable operation. Intangible assets represented by patents, human resources, copyrights and trademarks formed by enterprises consuming their own capital have fallen into a dilemma of idleness or devaluation due to shortage of funds. For such online game SMEs with valuable intangible assets such as technical patent and software copyright, if they can be supported by external credit providers, they can not only promote technological innovation in the game market, but also provide value-added premium for their external credit providers. Online game SMEs have professional and non-professional external credit suppliers in China: Professional credit suppliers include the four major state-owned banks and regulated third-party small and medium-sized financial service institutions in China; non-professional credit suppliers include private usury. The non-professional credit supplier also includes the non-professional credit supply of professional financial institutions, such as Baoshang Bank.

In 1970, Akerlof emphasized the consequences of asymmetry information in the market and the mechanism for solving asymmetry information in "The market for lemons: quality uncertainty and the market mechanism", which became the classic theory of information economics in modern microeconomics [1]. For credit-related information asymmetry issues, optimal credit guarantee ratio is influenced by three primary factors: government policy, macroeconomic conditions, and bank behavior [2]. Heterogeneous characteristics such as industry, corporate culture, and corporate organizational structure are also important factors affecting credit information asymmetry [3]. China's credit system environment is still insufficient, and the issue of credit imbalance has been discussed and studied for a long time [4]. In the process of changing social mobility from weak to strong, the Chinese government has adopted some new solutions to deal with the "credit dilemma", such as establishing a monitoring mechanism for corporate dishonesty [5, 6]. The timely exposure of credit information on social networks alleviates credit risk [7, 8]. Banks are an important supplier of external credit for SMEs, previous studies have pointed out that financial statement loans, mortgage-backed loans and credit scoring techniques are not applicable to small and medium-sized enterprises to a certain extent in the classification of bank credit supply information screening mechanisms [9, 10]. Relationship-based loans are more suitable for SMEs [11]. In addition to paying attention to financial information of financing demanders, relationship-based loans will also obtain information about intangible assets of financing demanders, such as suppliers and customer resources, through long-term contacts [12, 13]. The establishment of credit information sharing platform for bank loans can reduce its information asymmetry, thereby reducing the risk of corporate default [14]. Standardized relational financing is helpful for banks to lend to high-tech enterprises, but a revenue-sharing cooperation mechanism and a reward-penalty mechanism should be established between banks and enterprises [15, 16].

Government regulation is a corrective measure to solve monopoly [17]. Government regulation is significantly influenced by the industrial environment and network effects within the realm of SMEs operating in the online gaming sector. Although there is a significant positive effects of government regulation on the network industry with scale effects and high-tech characteristics, it concurrently fosters the emergence of market monopolies. The observed market monopoly displays characteristics of a long incubation period with human capital as the cost structure of special assets, the attributes of quasi-public goods of high-tech achievements, the spillover effect from R&D activities in the network industry, and the inertia of network consumption [18–20]. These characteristics not only increase the barriers to entry of innovative products, but also increase the difficulty for the online game industry to achieve full market

competition through industry self-regulation. The government should provide appropriate regulations to deal with the above situation [21]. Relevant government regulations have the functions of reducing market failures, reducing government support ineffectiveness, fostering values and orderly organizing industry for online game SMEs [22–24]. However, according to the regulatory capture theory, the imposing government regulation is subject to government regulation capture, it only increases profits rather than welfare output of the industry [25, 26]. The government should implement different and forward-looking support policies for different development stages of this industry [27–29]. The new impetus behind forward-looking policies not only benefits from legislative breakthroughs, but also benefits from the use and diffusion of social credit.

The shortcomings of the current research based on reading and analysis of previous research: 1) Research on the credit of small and medium-sized enterprises is mostly limited to the relationship between banks and enterprises. With the changes in the credit environment and the application of the Internet, the external credit providers of online game SMEs are diversified and specialized, and it is necessary to further understand these entities; 2) Insufficient identification of credit information for online game SMEs based on asset-light and knowledge-intensive features; 3) Lack of analysis from the perspective of revenue sharing mechanism on the credit issues of online game SMEs.

This article emphasizes the importance of corporate intangible asset information on the basis of corporate disclosing financial information. Accordingly, this paper constructs an evolutionary analysis framework of a cluster innovation network under the given fairness preferences. And build a dynamic game matrix between online game SMEs and credit providers with a revenue sharing model. Then conduct research on online game SMEs and their external credit providers. The basic logic method and research path diagram of this article is shown in Fig 1

The article is divided into the following sections: The first is the research on the status quo of online game SMEs. This part is divided into index construction and research methods, and multiple linear regression methods. The second is the dynamic game mechanism between credit suppliers and online game SMEs. The game mechanism is based on the discrimination of non-financial indicators and financial indicators of the credit suppliers. The third is to use the form of numerical simulation to simulate the behavioral characteristics of credit suppliers and online game SMEs under different credit synergistic benefit mechanisms, different

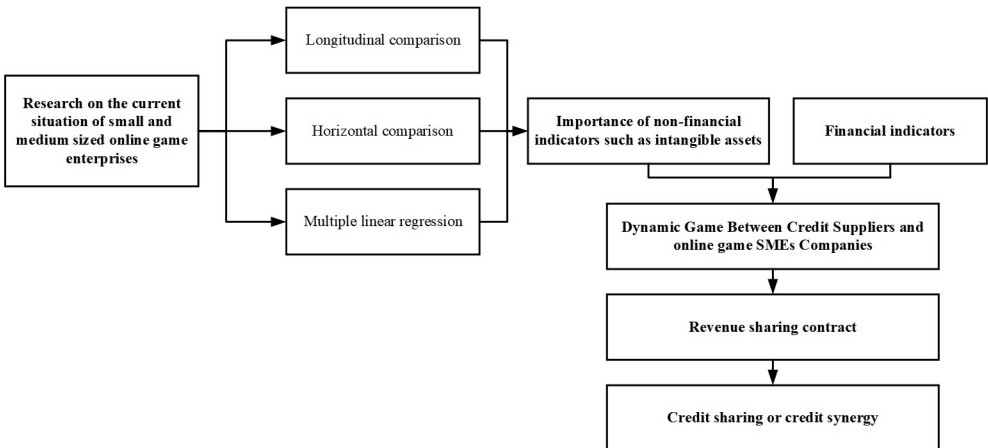

**Fig 1. The basic logic method and research path diagram of this article.**

revenue sharing methods, and different government penalty costs. The fourth is conclusions and policy recommendations. It especially supplements the important influence of non-financial indicators such as intangible assets obtained by the two main analysis methods-index construction research method and multiple linear regression method on the sustainable operation of online game SMEs.

The article is divided into the following four sections: 1) The research on the status quo of online game SMEs. This part is divided into index construction and research methods, and the process of applying multiple linear regression methods. 2) The dynamic game mechanism between credit suppliers and online game SMEs. The game mechanism is based on the discrimination of non-financial indicators and financial indicators of the credit providers. 3) Adopting the form of numerical simulation to simulate the behavioral characteristics of credit providers and online game SMEs under different credit synergistic benefit mechanisms, different revenue sharing methods, and different government penalty costs. 4) Conclusions and policy recommendations. It especially supplements the important influence of non-financial indicators such as intangible assets obtained by the two main analysis methods-index construction research method and multiple linear regression method on the sustainable operation of online game SMEs.

In view of the importance of non-financial indicators such as intangible assets and other important financial indicators can be used as important information for credit identification of credit providers. In the process of identifying all the important information of online game SMEs, credit providers engage in dynamic games with them due to the opportunistic tendency of online game SMEs to disclose information. In order to stabilize the results of the dynamic game between them, the credit providers can identify the high or low quality of the non-financial indicators and financial indicators of the SME, then sign a revenue sharing contract with the online game SMEs. Under this path, a credit sharing or credit coordination mechanism between credit suppliers and online game SMEs can be established to support cooperation.

## Data & methods

### Status of information disclosure of intangible assets of online game SMEs

The National Equities Exchange and Quotations(NEEQ) is a Chinese over-the-counter system for trading the shares of a public limited company. The listed companies are all high-tech companies and are different from the delisted companies in the original transfer system and the original STAQ and NET system listed companies. As of 2020, there are a total of 140 samples of online game SMEs in NEEQ. The basic status of online game SMEs is shown in Table 1.

### Longitudinal comparison of intangible asset information index

Due to the availability of data, this article selects five types of intangible assets of online game SMEs: trademarks, work copyrights, software copyrights, qualifications and patents. The index construction method is as follows: According to whether there are such intangible assets in the information disclosed by all sample companies from 2013 to 2019, the existence is 1 and the non-existence is 0. Add up the number of intangible assets in 7 years and divide by 7 to get the

**Table 1. The status quo of online game SMEs on the NEEQ.**

| Type of enterprise | non-ST and non-delisting | delisting | ST | ST and delisting | total |
|---|---|---|---|---|---|
| Type distribution | 67 | 51 | 22 | 1 | 140 |

Note: The sources of table are derived from Wind database.

**Table 2. Longitudinal comparison of intangible asset information index.**

| Section \ Classify | Information index of non ST and non-delisted intangible assets (67) | Information index disclosure of delisted intangible assets (51) | ST intangible assets information index disclosure (22) |
|---|---|---|---|
| [0,1.5] | 3 | 8 | 6 |
| (1.5,2] | 15 | 13 | 6 |
| (2,2.5] | 18 | 12 | 2 |
| (2.5,3] | 16 | 10 | 4 |
| (3,3.5] | 7 | 4 | 2 |
| (3.5,4] | 6 | 1 | 0 |
| (4,4.5] | 1 | 0 | 0 |
| (4.5,5] | 1 | 3 | 2 |
| (1.5,3] Proportion of quantity | 0.628 | 0.583 | 0.429 |

longitudinal index of the company's disclosure of the existence of various intangible assets in the past 7 years. The five types of intangible asset information indexes are summed up to obtain a comprehensive index, which is then compared and analyzed among enterprises. Table 2 shows the relevant results:

## Horizontal comparison of intangible asset information index

The horizontal comparison of the intangible asset information index is to divide the vertical index of all types of intangible assets of all enterprises by the number of different types of enterprises to obtain the average value of each type of enterprise, and then add the average value of the various indexes to obtain the average value of the composite index. Compare and analyze among enterprises of various classifications, as shown in Table 3:

The longitudinal and horizontal comparative research on the disclosure of intangible asset information index by enterprises of different classifications found that: The disclosure of intangible asset information of delisted and ST online game SMEs for 7 consecutive years (2013–2019) is not much different from that of companies that are neither ST nor delisted; In the horizontal comparative analysis, the delisted and ST companies are generally worse than the companies that are neither ST nor delisted.

**Table 3. Horizontal comparison of intangible asset information index.**

| average value \ Classify | Information index of non ST and non-delisted intangible assets (67) | Information index disclosure of delisted intangible assets (51) | ST intangible assets information index disclosure (22) |
|---|---|---|---|
| Average value of trademark index | 0.840 | 0.728 | 0.740 |
| Average value of copyright index of works | 0.313 | 0.244 | 0.234 |
| Average value of software copyright index | 0.945 | 0.868 | 0.838 |
| Average value of qualification index | 0.264 | 0.305 | 0.221 |
| Average patent index | 0.192 | 0.160 | 0.130 |
| Average of composite index | 2.554 | 2.305 | 2.162 |

This shows that online game SMEs that have been delisted and have delisting risks still have a large number of intangible assets, and the quality and value of this part of the asset need to be further explored.

## Linear regression of intangible assets and financial indicators

For online game SMEs, important financial indicators include "current liabilities", "operating cash inflow" and "operating income". These three indicators reflect the short-term external credit, cash flow and profitability of entities, can be used as the explained variable. In order to observe whether the intangible assets of online game SMEs are related to corporate credit and profitability, the number of trademarks, work copyrights, software copyrights, qualifications and patents are used as explanatory variables, and the scale, time effect and individual effect of the company are controlled. The setting of variables and the definition of indicators are shown in Table 4:

The results of descriptive statistics are shown in Table 5:

The empirical model uses a dual fixed effects model, as follows:

$$lnlb_{i,t} = \alpha_0 + \alpha_1 lnsb_{i,t} + \alpha_2 lnzp_{i,t} + \alpha_3 lnrj_{i,t} + \alpha_4 lnzz_{i,t} + \alpha_5 lnzl_{i,t} + \alpha_6 lnA_{i,t} + \varepsilon_{1i,t} + \mu_{1i,t} \quad (1)$$

$$lnca_{i,t} = \beta_0 + \beta_1 lnsb_{i,t} + \beta_2 lnzp_{i,t} + \beta_3 lnrj_{i,t} + \beta_4 lnzz_{i,t} + \beta_5 lnzl_{i,t} + \beta_6 lnA_{i,t} + \varepsilon_{2i,t} + \mu_{2i,t} \quad (2)$$

$$lnys_{i,t} = \gamma_0 + \gamma_1 lnsb_{i,t} + \gamma_2 lnzp_{i,t} + \gamma_3 lnrj_{i,t} + \gamma_4 lnzz_{i,t} + \gamma_5 lnzl_{i,t} + \gamma_6 lnA_{i,t} + \varepsilon_{3i,t} + \mu_{3i,t} \quad (3)$$

In formulas (1), (2) and (3), i and t are the sample company and year respectively, $\alpha_0$, $\beta_0$, and $\gamma_0$ are constant terms, and $\varepsilon_{1i,t}$, $\varepsilon_{2i,t}$, and $\varepsilon_{3i,t}$ are the year fixed effects of various formulas, respectively. $\mu_{1i,t}$, $\mu_{2i,t}$, $\mu_{3i,t}$ are the random error terms of various formulas respectively.

Tables 6 to 8 show that the existence of at least one type of intangible asset among the five types of intangible assets has a significant promotion effect on the external short-term credit, cash inflows and operating income of non-ST and non-delisted companies, ST and delisted companies. In terms of external short-term credit, cash inflows and profitability, non-ST and non-delisted companies do perform better than ST and delisted companies in terms of trademarks and qualification-type intangible assets, the trademarks and patents of ST and delisted companies still have value for further research.

In the classified samples, trademarks in the non-ST and non-delisted samples not only significantly promote the increase of external credit and operational cash inflow, but also increase

**Table 4. Setting and definition of variables.**

| Indicator type | Indicator name | Symbol | Definition |
|---|---|---|---|
| Explained variables | Current liabilities | lnlb | The value of current liabilities is logarithmic. |
| | Operating cash inflow | lnca | The value of operating cash inflow is logarithmic. |
| | Operating income | lnys | The value of operating income is logarithmic. |
| Explanatory variables | Trademark | lnsb | Add 1 to the number of trademarks and take the logarithm. |
| | Work copyright | lnzp | The number of copyrights of the work is increased by 1 and then taken as the logarithm. |
| | Software copyright | lnrj | Add 1 to the number of software copyrights and take the logarithm. |
| | Qualification | lnzz | The number of qualifications is increased by 1 and the logarithm is taken. |
| | Patent | lnzl | The number of patents is increased by 1, and then the logarithm is taken. |
| Control variables | Total assets | lnA | The total asset value takes the logarithm. |
| | Year | year | Year, from 2013 to 2019. |

**Table 5. Descriptive statistics of variables.**

| Variables | Mean | Standard deviation | Minimum | Max | Observations |
|---|---|---|---|---|---|
| lnlb | 15.79521 | 1.680201 | 8.919908 | 21.35056 | 917 |
| lnca | 17.24087 | 1.545883 | 12.24381 | 22.45112 | 917 |
| lnys | 17.02959 | 1.829473 | 0 | 22.27898 | 917 |
| lnsb | 2.391095 | 1.712208 | 0 | 6.622736 | 917 |
| lnzp | 0.6507322 | 1.263351 | 0 | 6.472346 | 917 |
| lnrj | 2.668488 | 1.263875 | 0 | 5.545177 | 917 |
| lnzz | 0.2868222 | 0.5205186 | 0 | 3.401197 | 917 |
| lnzl | 0.3575386 | 0.9552377 | 0 | 5.78996 | 917 |
| lnA | 17.31407 | 1.524684 | 10.92044 | 21.96863 | 917 |

the operating income of such online game SMEs(Table 7). Therefore, the trademark quality of these online game SMEs that have not been delisted or ST is relatively good, and the intangible assets of the trademark can become an important signal of whether the enterprise can attract more external credit. Another important indicator of intangible assets is the qualification of such enterprises. If the number of such enterprises is more, it will promote the increase of credit, operating cash inflow and operating income of online game SMEs. Therefore, the intangible assets of qualification have also become an important indicator of whether the online game SMEs of this classification have the sustainable management ability.

**Table 6. Full sample empirical results.**

| Explained variables | Lnlb | Lnca | lnys |
|---|---|---|---|
| lnsb | 0.145*** | 0.0413* | 0.0339 |
|  | (0.0285) | (0.0221) | (0.0301) |
| lnzp | -0.0317 | -0.0442 | -0.0873** |
|  | (0.0348) | (0.0270) | (0.0367) |
| lnrj | -0.133*** | -0.0263 | -0.0551 |
|  | (0.0402) | (0.0312) | (0.0424) |
| lnzz | 0.0882 | 0.186** | 0.348*** |
|  | (0.0941) | (0.0730) | (0.0992) |
| lnzl | 0.0827* | -0.00273 | -0.0413 |
|  | (0.0461) | (0.0358) | (0.0486) |
| lnA | 0.772*** | 0.837*** | 0.926*** |
|  | (0.0303) | (0.0235) | (0.0320) |
| Individual effect | Controlled | Controlled | Controlled |
| Time effect | Controlled | Controlled | Controlled |
| Constant | 2.527*** | 2.904*** | 1.150** |
|  | (0.484) | (0.375) | (0.510) |
| Observations | 917 | 917 | 917 |
| $R^2$ | 0.550 | 0.680 | 0.579 |
| $Adj - R^2$ | 0.5443 | 0.6761 | 0.5729 |

Note: Standard errors are in parentheses,

*** p<0.01,

** p<0.05,

* p<0.1.

**Table 7. Empirical results of non-ST and non-delisting samples.**

| Explained variables | lnlb | Lnca | lnys |
|---|---|---|---|
| lnsb | 0.208*** | 0.0543* | 0.105** |
| | (0.0402) | (0.0308) | (0.0451) |
| lnzp | -0.0172 | -0.0182 | -0.0921* |
| | (0.0483) | (0.0369) | (0.0542) |
| lnrj | -0.187*** | -0.0174 | -0.131* |
| | (0.0645) | (0.0493) | (0.0724) |
| lnzz | 0.320** | 0.383*** | 0.772*** |
| | (0.142) | (0.109) | (0.160) |
| lnzl | -0.0487 | -0.0603 | -0.146** |
| | (0.0620) | (0.0475) | (0.0696) |
| lnA | 0.787*** | 0.862*** | 1.003*** |
| | (0.0480) | (0.0367) | (0.0539) |
| Individual effect | Controlled | Controlled | Controlled |
| Time effect | Controlled | Controlled | Controlled |
| Constant | 2.026*** | 2.302*** | -0.223 |
| | (0.760) | (0.582) | (0.854) |
| Observations | 469 | 469 | 469 |
| $R^2$ | 0.542 | 0.678 | 0.565 |
| $Adj - R^2$ | 0.5296 | 0.6694 | 0.5539 |

Notes:

\* $P<0.01$,

\*\* $P<0.05$,

\*\*\* $P<0.001$; t-statistics are in parentheses.

In the classified samples, the relationship between the five types of intangible assets and current liabilities, operating cash inflow and operating income of Internet small and medium-sized game enterprises in ST and delisted samples (Table 8). This paper finds that although trademarks in this category increase current liabilities, they do not increase operating cash inflow and operating income, so the value of trademarks in this category is limited to short-term external credit increase. At the same time, the patent intangible assets of the classified enterprises not only promote the increase of current liabilities, but also increase the operating cash flow and operating income of enterprises. However, the reason of delisting and ST of such enterprises may be that their qualifications are inferior to those of enterprises without ST and delisting, and the copyright of their works and software copyright cannot significantly promote the increase of current liabilities, operating cash inflow and operating income.

However, some intangible assets of the delisting and ST Internet small and medium-sized game enterprises still have some significant promoting functions. If external credit providers ignore the value of these intangible assets, then these market players, who could continue to operate and provide innovation power for the market, will be eliminated, thus reducing the technological innovation capacity of the market.

Regardless of the longitudinal and horizontal comparison of related intangible asset indicators, or the analysis of the regression results of the total sample and sub-samples, according to the regression results of Tables 6 to 8, the regression results show that, the promotion functions related to intangible assets of normal operating online game SMEs and ST and delisted all exist, but there are differences in the promotion effect of different types of intangible assets.

**Table 8. Empirical results of ST and delisting samples.**

| Explained variable | lnlb | Lnca | lnys |
|---|---|---|---|
| lnsb | 0.115*** | 0.0277 | -0.0386 |
| | (0.0388) | (0.0329) | (0.0385) |
| lnzp | -0.102** | -0.0913** | -0.0685 |
| | (0.0508) | (0.0432) | (0.0505) |
| lnrj | -0.0498 | -0.0106 | 0.0369 |
| | (0.0495) | (0.0421) | (0.0492) |
| lnzz | -0.204* | 0.0179 | -0.0299 |
| | (0.122) | (0.103) | (0.121) |
| lnzl | 0.346*** | 0.143** | 0.131* |
| | (0.0717) | (0.0609) | (0.0712) |
| lnA | 0.716*** | 0.769*** | 0.821*** |
| | (0.0407) | (0.0346) | (0.0405) |
| Individual effect | Controlled | Controlled | Controlled |
| Time effect | Controlled | Controlled | Controlled |
| Constant | 3.544*** | 4.096*** | 2.951*** |
| | (0.650) | (0.552) | (0.645) |
| Observations | 420 | 420 | 420 |
| $R^2$ | 0.567 | 0.639 | 0.594 |
| $Adj - R^2$ | 0.5541 | 0.6283 | 0.5823 |

Notes:

* P<0.01,

** P<0.05,

*** P<0.001; t-statistics are in parentheses.

The conclusion shows that it is necessary to conduct a penetrating identification analysis of these ST and delisted online game SMEs in order to better provide credit support for these enterprises that should be sustainable but delisted due to the exhaustion of funds. Among them, intangible assets are important identification objects.

## Dynamic game mechanism under credit synergy

### Credit capacity and credit supply

The game mechanism between online game SMEs and credit providers lies in effective information disclosure. The effective information includes corporate financial indicators and other non-financial indicators [28]. Online game SMEs and credit providers play games based on the company's disclosed information and market information, and online game SMEs belong to the financing demand side. Empirical research results show that online game SMEs have some valuable intangible assets, but these valuable information or assets are not recognized by credit providers. This article believes that in addition to financial indicators, intangible assets are also an important indicator of credit ability identification. And divide the credit supplier into professional credit supplier and non-professional credit supplier, and divide online game SMEs into high-credit ability enterprises and low-credit ability enterprises. Among them, credit suppliers include governments, banking institutions, non-bank financial institutions, large investment institutions, and large enterprises that endorse the credit of SMEs. These entities mainly create credit supply and have different lending mechanisms and have different

credit risk expectations. Non-professional financial institutions such as private usury are correspondingly regarded as non-professional credit providers. In addition, the non-professional credit behavior of professional credit suppliers is also classified as non-professional credit providers such as Baoshang Bank.

Online game SMEs with non-professional credit behavior and low credit capabilities pretend to be high-credit companies in order to obtain opportunistic benefit behavior and rent-seeking behavior, such as obtaining government subsidies or take fraudulent behavior after the credit supply of professional credit providers, such behaviors are regulated by government and the relevant regulatory authorities. Therefore, non-professional credit providers and online game SMEs that pretend to be high-credit capacity are the main targets of government regulation and impose penalties on their non-compliant financing behaviors. Regulatory authorities use the punitive cost for the construction of professional and high-credit capability entities, that is, the penalty costs of non-professionals and disguised as high-credit capability entities make up for the losses of professional and high-credit capability entities.

## Game payoff matrix

Assuming that in the market, the probability of the credit provider taking professional behavior is $y$, and the probability of online game SMEs choosing high credit capability is $x$. The professional credit provider supports loan to online game SMEs with high credit capabilities, and the high-credit financing demand side obtains benefit $R_1$ minus the cost $c_1$ that the financing demand side pays to become a high-credit enterprise. At the same time, high-credit enterprises and professional credit providers form a credit synergy model based on intangible assets (trademarks, work copyrights, software copyrights, qualifications, patents, etc.). Credit synergy refers to the relationship in which the credit supplier provides credit to the financing demander based on the credit capability of the financing demander, and forms a credit relationship with the financing demander such as technology complementarity, capability complementarity, resource complementarity, and demand complementarity. This paper takes intangible asset trademarks, work copyrights, software copyrights, qualifications and patents as important information for identifying the creditworthiness of Internet small and medium-sized game companies. $U$ is the overflow income generated by the establishment of a credit synergy relationship between the high-credit ability enterprise and the professional credit provider based on intangible assets. The high-credit ability enterprise shares the overflow income as $\omega U$ and the professional credit provider shares the overflow income as $(1 - \omega)U$. This part $\omega U$ should be added to the total credit benefit of a high-credit ability enterprise to obtain a professional credit provider, and its total benefit is $(R_1 + \omega U - c_1)$, $\omega \in [0, 1]$. Similarly, the benefit obtained by the professional credit provider is $L_1$ minus the cost $c_2$ paid by the credit supplier to become a professional credit provider plus the shared overflow income part $(1 - \omega)\,U$, and then its total benefit is $[L_1 + (1 - \omega)U - c_2]$; The credit provider becomes a non-professional credit provider with probability $(1 - y)$ (the non-professional credit provider does not require additional costs), and provides credit to high-credit companies. The benefit $R_1$ obtained by high-credit enterprises is increased by the government's subsidy $k$ to high-credit enterprises for obtaining low-quality credit (this part of government subsidies can be set as government subsidies derived from the penalty cost of non-professional credit providers), subtract the effort cost $c_1$ of online game SMEs to become high-credit companies, the total benefit obtained by the high-credit ability enterprise is $(R_1 + k - c_1)$, and the total benefit obtained by the non-professional credit provider is $(L_1 + e_2 - k)$, where $e_2$ is the benefit obtained by the non-professional credit provider due to negative opportunism, $k$ is the punishment for the credit provider to become a non-professional credit provider or to provide non-professional credit. Further

clarification is needed regarding government subsidies $k$, the government's subsidy targets for large-scale online game companies and online game SMEs are different, which can be roughly divided into the following two categories: First, take Tencent Games, a large online game enterprise, as an example. The government subsidies for such large enterprises focus on their sustainable operating space, sustainable operating income, and sustainable taxation, with implicit government subsidies such as preferential land use, etc. policy; Second, the government's subsidies to online game SMEs focus more on the positioning of the innovation, employment, and industry learning and training subjects of online game SMEs. The main research object of this paper is online game SMEs, so government subsidies are more focused on the second category.

The probability that a online game SME is a low-credit company is $(1 - x)$, and the comprehensive benefit of obtaining credit from a professional credit provider is $(R_1 + e_1 - k)$, and $k$ is the penalty for a low-credit company disguising as a high-credit company. In order to simplify the calculation, it is assumed that the penalty cost of online game SMEs with low credit capabilities and the penalty cost of non-professional credit providers are both $k$, and $e_1$ is the opportunistic benefit of online game SMEs becoming low-credit companies obtaining professional credit. The comprehensive benefit obtained by the credit provider is $(L_1 + k - c_2)$, in which the penalty for the non-professional credit supplier $k$ is subsidizing the professional credit provider; the credit obtained by the low-credit ability enterprise from the non-professional credit provider is $R_1$, the benefit of non-professional credit providers who provide credit to enterprises with low credit ability is $L_1$. Among them, $x, y \in [0, 1]$ and the probability is also a function of time $t$, the probability of different types of credit providers in the market and online game SMEs will change as time changes. And suppose that the government penalty cost and government subsidy are both $k$, the cost of the credit supplier's efforts to become a professional credit provider $c_2$, and the cost $c_1$ (including opportunity cost) of the online game SMEs' efforts to become a high-credit company. The three satisfying the inequality relationship $k < c_1$ and $k < c_2$ means that the cost of online game SMEs striving to become high-credit companies is greater than the supply costs of non-professional credit providers and the penalty costs of low-credit companies. In the same way, the cost of a credit provider's efforts to become a professional credit provider is greater than the penalty costs or government subsidies for non-professional credit providers and online game SMEs with low credit capabilities. The game matrix is shown in Table 9:

## The evolution process of the equilibrium point

Based on the game matrix above, the game strategy between online game SMEs and credit providers is suitable to be described through the dynamic replication process in the evolutionary game theory [30]. Therefore, a dynamic replication equation between the credit provider and online game SMEs can be constructed to describe the evolution process of the strategies of the two parties in the game. The key for the two parties in the game to choose between professional

**Table 9. Payment matrix for credit matching.**

| Game and decision | | Credit supplier | |
|---|---|---|---|
| | | **Professional** <br> **($y$)** | **Non-professional** <br> **($1 - y$)** |
| Online game SMEs | High credit ability ($x$) | $(R_1 + \omega U - c_1, L_1 + (1 - \omega) U - c_2)$ | $(R_1 + k - c_1, L_1 + e_2 - k)$ |
| | Low credit worthiness <br> ($1 - x$) | $(R_1 + e_1 - k, L_1 + k - c_2)$ | $(R_1, L_1)$ |

and non-professional, high-credit ability and low-credit ability type enterprises lies in their own pursuit of maximizing their own benefits.

The benefit expectation function of online game SMEs choosing to become high-credit companies is shown in (4):

$$\pi_{11} = y(R_1 + \omega U - c_1) + (1 - y)(R_1 + k - c_1) \tag{4}$$

The benefit expectation function of online game SMEs choosing to become low-credit companies is shown in (5):

$$\pi_{12} = y(R_1 + e_1 - k) + (1 - y)R_1 \tag{5}$$

Then the average benefit expectation function of online game SMEs is shown in (6):

$$\pi_1 = x\pi_{11} + (1 - x)\pi_{12} \tag{6}$$

Therefore, the dynamic replication equation for whether online game SMEs become high-credit and low-credit companies is shown in (7):

$$\begin{aligned} f_1(x, y) &= dx/dt = (\pi_{11} - \pi_1) = x(1 - x)[\pi_{11} - \pi_{12}] \\ &= x(1 - x)[y(\omega U - e_1) + k - c_1] \end{aligned} \tag{7}$$

In the same way, the average benefit expectation function of the credit provider is (8):

$$\pi_2 = y\pi_{21} + (1 - y)\pi_{22} \tag{8}$$

The dynamic replication equation of whether a credit provider becomes a professional or non-professional credit provider is (9):

$$\begin{aligned} f_2(x, y) &= dy/dt = (\pi_{21} - \pi_2)x = y(1 - y)[\pi_{21} - \pi_{22}] \\ &= y(1 - y)\{[(1 - \omega)Ux - xe_2] + k - c_2\} \end{aligned} \tag{9}$$

In the market, credit providers and online game SMEs engage in dynamic games, and credit providers seek SMEs with high credit capabilities in order to obtain value added; And online game SMEs seek professional credit providers to obtain credit support, thereby enhancing their operational capabilities. In the process of dynamic game between the two, there is an effective equilibrium point. According to the formula (10):

$$\begin{cases} f_1(x, y) = dx/dt = 0 \\ f_2(x, y) = dy/dt = 0 \end{cases} \tag{10}$$

And then get 5 equilibrium points, respectively: $O(0, 0)$, $A(0, 1)$, $B(1, 0)$, $C(1, 1)$, $D\left(\frac{c_2 - k}{[(1 - \omega)U - e_2]}, \frac{c_1 - k}{(\omega U - e_1)}\right)$, the five equilibrium points are the equilibrium points of the evolutionary game, and the boundary of the evolutionary solution domain of the two-agent game is: $\{(x, y)|0 \le x \le 1; 0 \le x \le 1\}$. According to the above obtained dynamic replication equation between online game companies SMEs and credit providers, the Jacobian matrix can be obtained as (11):

$$J = \begin{bmatrix} \partial f_1/\partial x & \partial f_1/\partial y \\ \partial f_2/\partial x & \partial f_2/\partial y \end{bmatrix} \tag{11}$$

The stability of the equilibrium point is judged by the sign of the trace $tr(J)$ of the matrix J and the sign of the determinant $det(J)$ to determine whether the equilibrium point belongs to

**Table 10. Equilibrium point and stability of the evolutionary game between online game SMEs and credit provider.**

| Equilibrium point | tr(J) | det (J) | Stability |
|---|---|---|---|
| $O(0,0)$ | $tr(J)<0$ | $det(J)>0$ | Stable point(ESS) |
| $A(0,1)$ | $tr(J)>0$ | $det(J)>0$ | Unstable |
| $B(1,0)$ | $tr(J)>0$ | $det(J)>0$ | Unstable |
| $C(1,1)$ | $tr(J)<0$ | $det(J)>0$ | Stable point(ESS) |
| $D\left(\frac{c_2-k}{[(1-\omega)U-e_2]}, \frac{c_1-k}{(\omega U-e_1)}\right)$ | $tr(J)=0$ | $det(J)<0$ | Saddle point |

the stable point *(ESS)*, unstable point and saddle point of the evolutionary game. The results of evolutionary game are shown in Table 10.

The stable point (ESS) of the evolutionary game between the credit provider and online game SMEs are $O(0,0)$, $C(1,1)$, the corresponding strategies are (low-credit enterprises, non-professional credit providers), (high-credit enterprises, professional credit providers); The instability points are $A(0,1)$ and $B(1,0)$, and the corresponding strategies are (low-credit ability enterprises, professional credit providers), (high-credit ability enterprises, non-professional credit providers); $D\left(\frac{c_2-k}{[(1-\omega)U-e_2]}, \frac{c_1-k}{(\omega U-e_1)}\right)$ is the saddle point, which lies on the plane range $\{(x,y)| 0 \le x \le 1; 0 \le x \le 1\}$.

The evolutionary equilibrium path of both parties is shown in Fig 2:

## Evolutionary game scenario simulation

The scenario simulation of the game between the two parties mainly considers the benefit sharing mechanism of the two parties' credit synergy, and the incentive and regulation of the participants are also taken into consideration. In the credit market, due to information asymmetry, credit supply entities and financing demanders are based on the premise of maximizing their respective interests. Capital allocation will choose between professional and non-

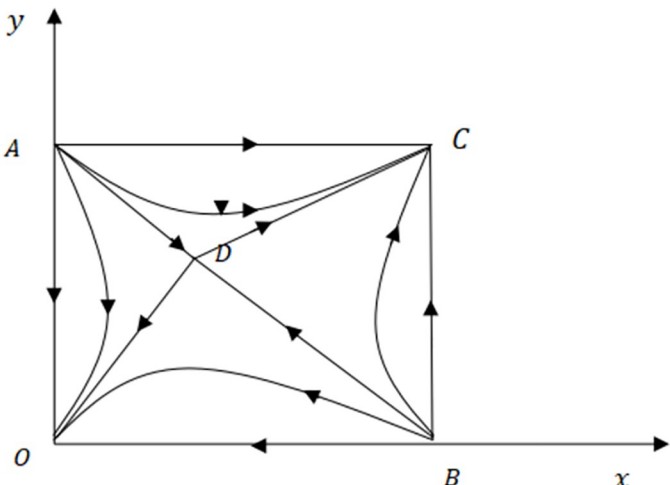

**Fig 2. The equilibrium point and saddle point of the evolutionary game between online game SMEs and credit providers.**

professional, high-credit ability and low-credit ability. The game between the two has become an important internal mechanism for the effective allocation of credit. In order to better represent the dynamic evolution behavior of the two, numerical simulation is used to describe.

The ideal situation is that professional credit providers and online game SMEs with high credit capabilities achieve credit synergy based on benefit sharing, and when the benefit sharing does not satisfy their respective interests' maximization, the credit synergy cannot be implemented. Set the time step as $\Delta t$, the probability that credit providers and online game SMEs become professional and non-professional subjects, and subjects with high credit ability and low credit ability are both 0.5. The size of the credit synergy benefit $U$ is represented by 0.5, 1, 1.5, and 2.5. 0.5 means that the return is not ideal, 1, 1.5 means that the return is relatively general, and 2.5 means that the benefit is good. The distribution coefficient $\omega$ of benefit sharing is represented by 0.2, 0.5, and 0.9 respectively, where 0.2 and 0.9 indicate unreasonable distribution, and 0.5 indicates relatively reasonable distribution. The punitive cost $k$ for online game SMEs with low credit capabilities and non-professional credit providers is 0.01, 0.02, 0.03, and 0.1. 0.01 represents a low penalty cost, 0.02 represents a general penalty cost, 0.03 and 0.1 represent a higher penalty cost, and other parameters are fixed values. The evolutionary game path of the two is shown in Figs 3–8.

## Scenario simulation of different credit synergies

Assuming related parameters $\omega = 0.2$; $e_1 = 0.3$; $e_2 = 0.3$; $k = 0.03$; $c_1 = 0.2$; $c_2 = 0.3$; Fig 3-1 is the simulation result of the credit synergistic benefit of online game SMEs. Under different credit synergies, online game SMEs show greater differences. Under relatively low credit synergies ($U = 0.5$, $U = 1$), online game SMEs tend to be low-credit companies. With relatively high credit synergies ($U = 1.5$, $U = 2.5$), online game SMEs tend to be high-credit enterprises. The lower the credit synergistic benefit leads to the shorter the time for online game SMEs to become low-credit capacity enterprises, the higher the credit synergistic benefit, the shorter the time for online game SMEs to become high-credit capacity enterprises. Similarly, as shown in the Fig 3-2, under different credit synergies, the performance of credit providers also has greater differences. Under lower credit synergies ($U = 0.5$, $U = 1$), the greater the probability of credit providers becoming non-professional. With higher credit synergies ($U = 1.5$, $U = 2.5$), the probability of a credit provider becoming a professional credit provider is also greater. And the lower the credit synergy benefit, the shorter the credit provider becomes a non-professional credit provider, and the higher the credit synergy benefit, the shorter the credit provider becomes a professional credit provider. By comparing the simulation behaviors of online game SMEs and credit providers, it is found that the credit providers are more sensitive to the behavioral feedback of online game SMEs.

Fig 4-1 and 4-2 further describe the situation where credit synergies are not ideal ($U = 0.5$) and the situation where credit synergies are ideal ($U = 1.5$). In the case of unsatisfactory credit synergies, the performance of credit providers is more sensitive than that of online game SMEs. That is, in the case of poor credit synergies, credit providers are more likely to become non-professional credit providers more quickly.

Under the ideal situation of credit synergies, the tendency of credit providers to become professional is not as fast as online game SMEs becoming high-credit. That is, online game SMEs show enthusiasm for ideal credit synergies (credit synergistic benefit is more conducive to online game SMEs).

The credit provider shows irritability to the behavior of online game SMEs, such as when credit synergistic benefit $U = 1.5$, the tendency of online game SMEs to tend to be high-credit

(1)

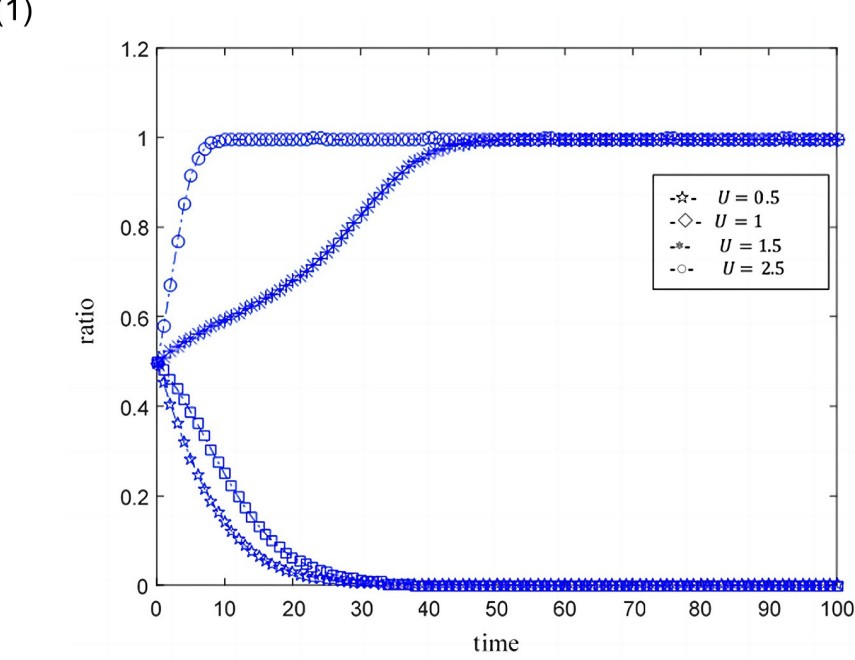

(2)

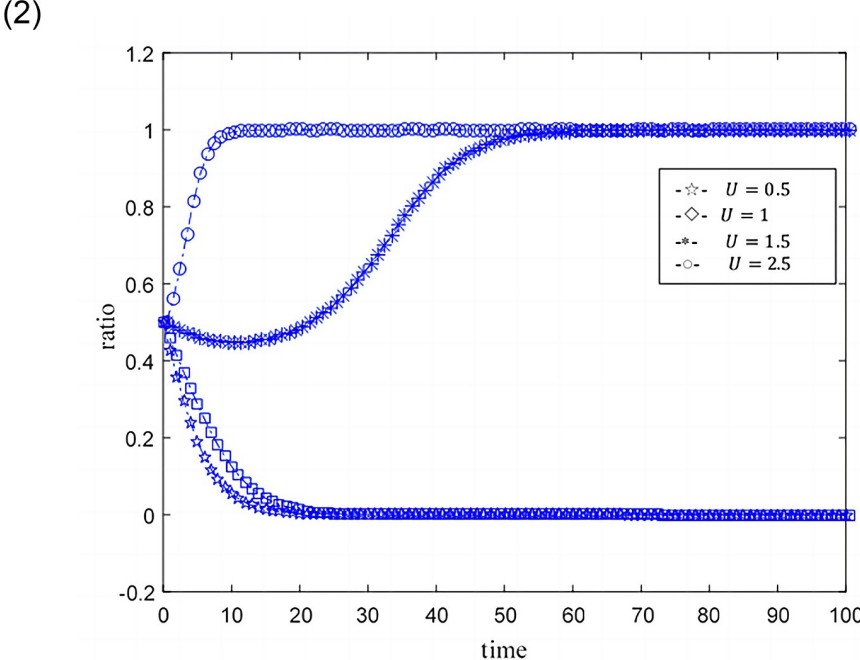

**Fig 3.** 1. Simulation behaviors of online game SMEs under different credit synergies. 2. Simulation behaviors of credit suppliers under different credit synergies.

(1)

(2)

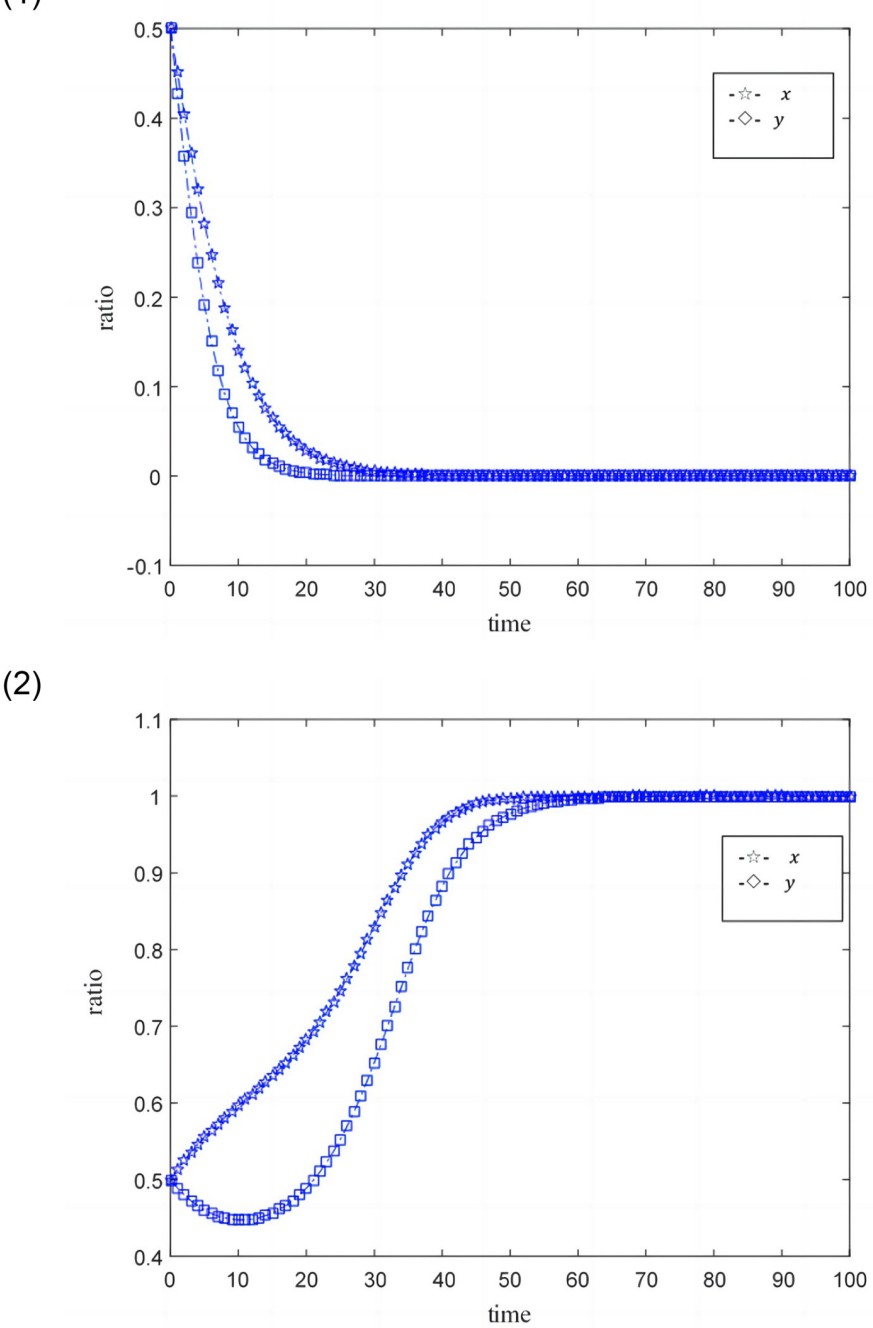

**Fig 4.** 1. Simulation results under unsatisfactory credit synergies ($U = 0.5$). 2. Simulation results under ideal credit synergies ($U = 1.5$).

companies has a tendency to increase first and then slow down, leading to the direct tendency of credit providers to be non-professional. After observing online game SMEs tend to become high-credit companies, the credit provider adjusted their behavior in a timely manner to the professional credit provider.

(1)

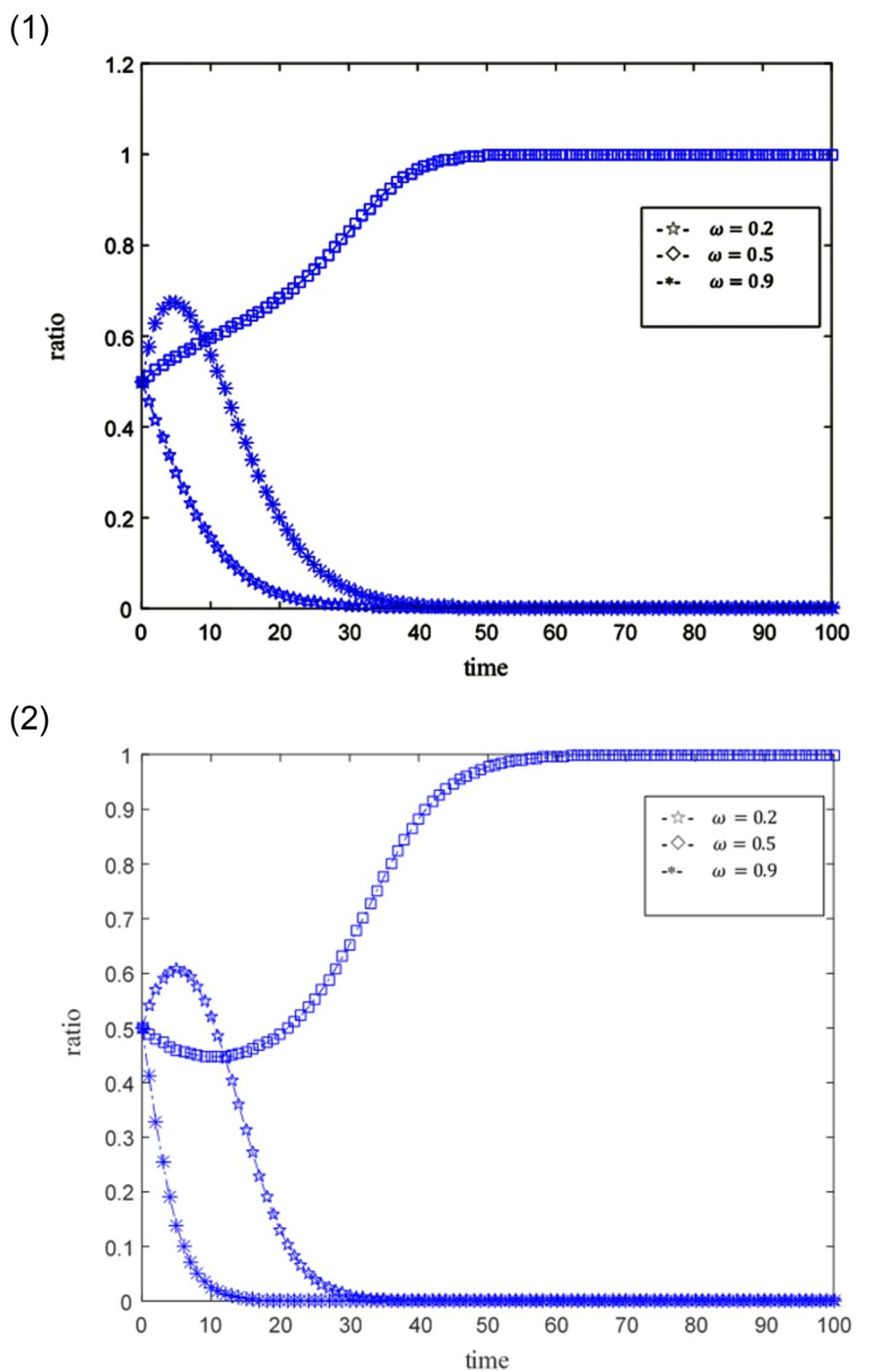

(2)

**Fig 5.** 1. Simulation behaviors of online game SMEs under different revenue sharing coefficients. 2. Simulation behavior of credit supply side under different revenue sharing coefficients.

## Scenario simulation of different benefit sharing coefficients

Fig 5 shows the simulation behavior of online game SMEs and credit providers under different benefit sharing coefficients. The relevant parameters are $U = 1.5$, $e_1 = 0.3$, $e_2 = 0.3$, $k = 0.03$, $c_1 = 0.2$, $c_2 = 0.3$. The specific characteristics of different benefit sharing coefficients are as

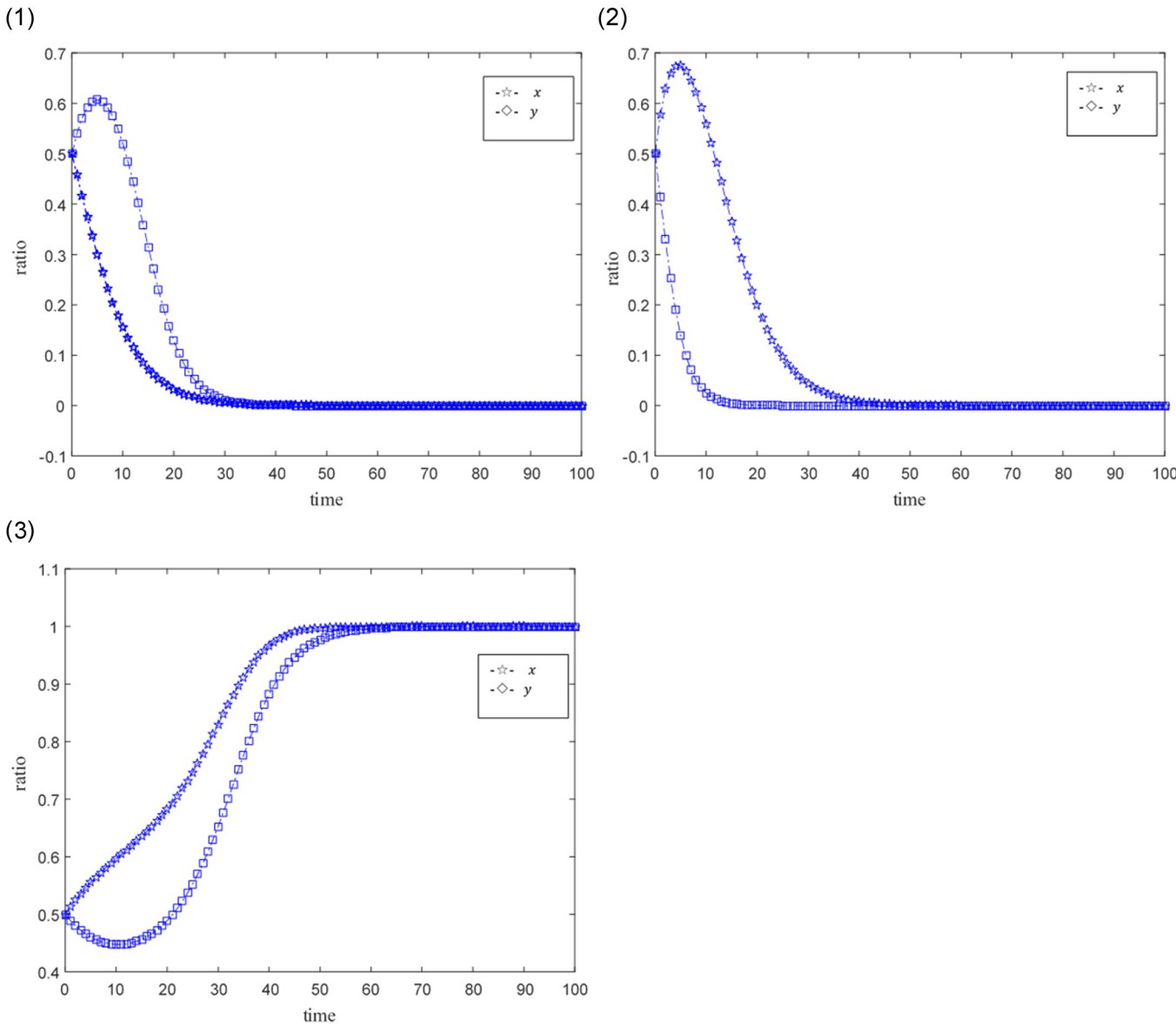

**Fig 6.** 1. Unreasonable income distribution ($\omega = 0.2$). 2. Unreasonable income distribution ($\omega = 0.9$). 3. Reasonable income distribution ($\omega = 0.5$).

follows: $\omega = 0.2$ is an unreasonable benefit sharing coefficient; $\omega = 0.5$ is a relatively reasonable revenue benefit coefficient; $\omega = 0.9$ is an unreasonable benefit sharing coefficient.

Fig 5-1 shows the simulation behavior of online game SMEs under different benefit sharing coefficients. Under unreasonable benefit sharing coefficients $\omega = 0.2$ and $\omega = 0.9$, online game SMEs tend to be low-credit companies. And between these two unreasonable sharing coefficients, $\omega = 0.2$ is even more unreasonable than $\omega = 0.9$ for online game SMEs. It shows that under the unreasonable benefit sharing coefficient, online game SMEs are more inclined to high benefit sharing ratio.

Fig 5-2 shows the simulation behavior of the credit provider under different benefit sharing coefficients. Under the unreasonable benefit sharing coefficients $\omega = 0.2$ and $\omega = 0.9$, the credit provider tends to transform into a non-professional credit provider. And between these two

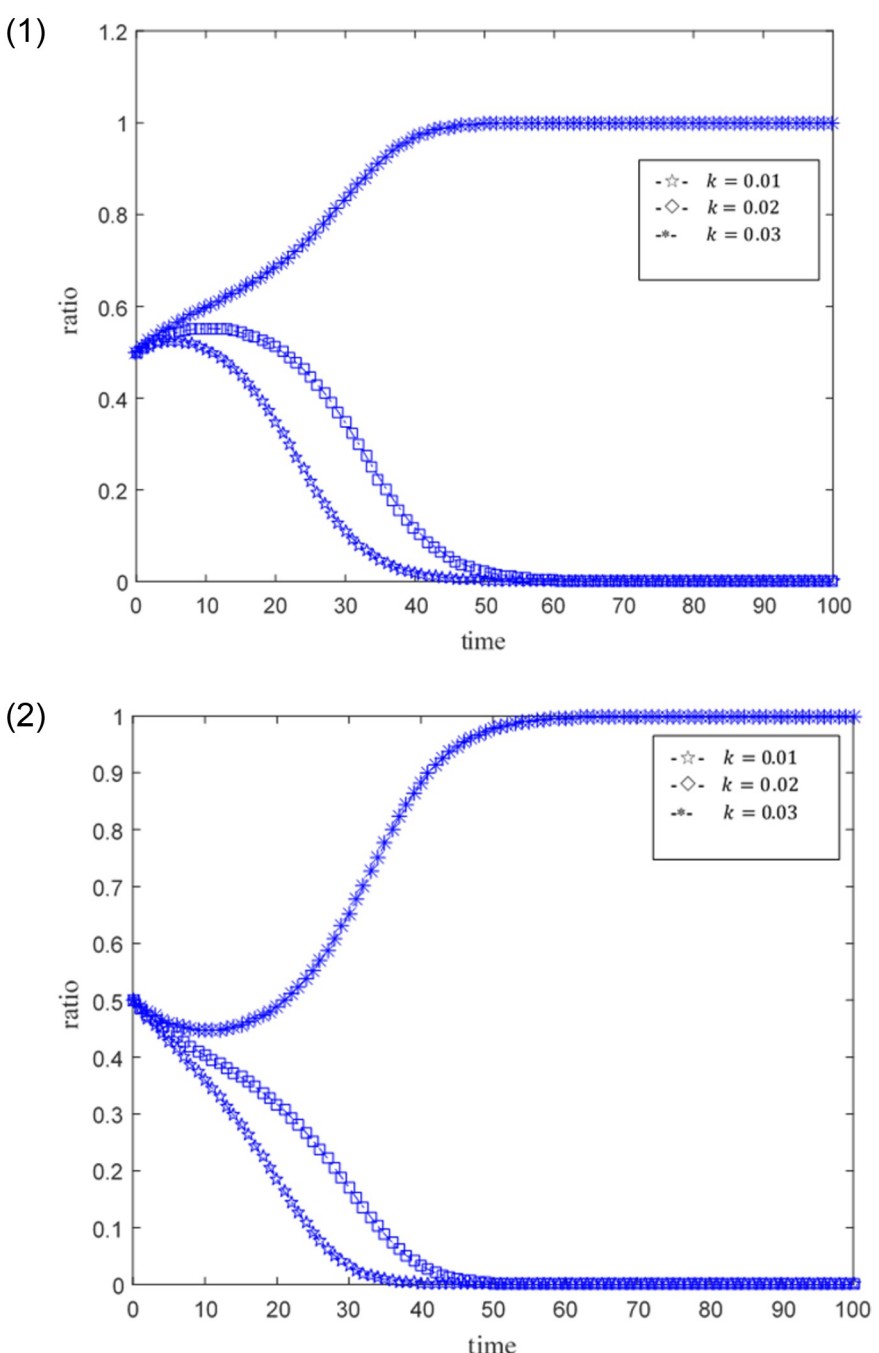

**Fig 7.** 1. Simulation behaviors of online game SMEs under different government penalty cost constraints. 2. Simulation behavior of credit suppliers under different government penalty cost constraints.

unreasonable sharing coefficients, $\omega = 0.9$ is even more unreasonable than $\omega = 0.2$ for the credit provider. It shows that under the unreasonable benefit sharing coefficient, the credit provider is more inclined to adopt a low benefit sharing ratio. Under a reasonable credit sharing benefit coefficient $\omega = 0.5$, online game SMEs and credit providers tend to be high-credit companies and professional credit providers. At this time, online game SMEs and credit

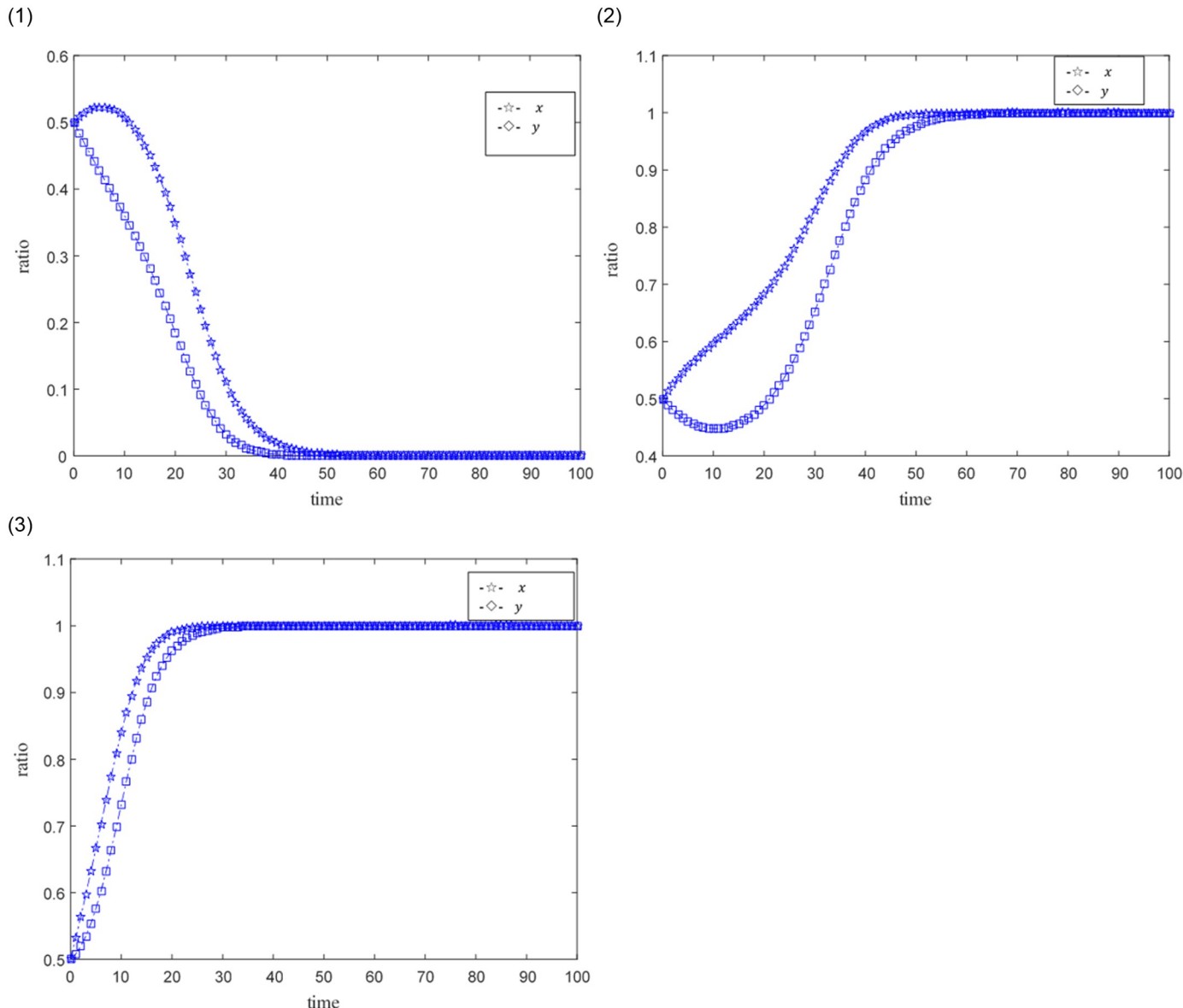

**Fig 8.** 1. Simulation results under low government penalty costs. 2. Simulation results under relatively high government penalty costs. 3. Simulation results under high government penalty costs.

providers are highly credit-worthy and professional credit subjects have relatively considerable benefits.

Fig 6 shows other relevant parameters such as $U = 1.5$, $e_1 = 0.3$, $e_2 = 0.3$, $k = 0.03$, $c_1 = 0.2$ and $c_2 = 0.3$ under the three benefit distribution coefficients. Comparison of simulation behaviors between online game SMEs and credit providers, when the benefit distribution coefficient $\omega = 0.2$ (Fig 6-1) is unreasonable, online game SMEs are more likely to turn into low-credit companies. When the credit distribution is unreasonable coefficient $\omega = 0.9$ (Fig 6-2), the credit provider is more likely to transform into a non-professional credit provider. When credit allocation is reasonable coefficient $\omega = 0.5$ (Fig 6-3), online game SMEs and credit providers tend to transform into credit entities with high credit ability and professionalism.

However, online game SMEs have shown greater enthusiasm, and credit providers have shown corresponding follow-up.

## Scenario simulation of different government penalty cost constraints

Fig 7 shows the simulation behaviors of online game SMEs and credit providers under different government penalty cost constraints. Assuming that the relevant main parameters are set to $\omega = 0.5$, $U = 1.5$, $e_1 = 0.3$, $e_2 = 0.3$, $c_1 = 0.2$ and $c_2 = 0.3$, simulations of constraint scenarios under different penalty costs are performed respectively.

Fig 7-1 shows the simulation behavior of online game SMEs under the constraints of different government penalty costs. When the government penalty cost is lower, such as $k = 0.01$ and $k = 0.02$, online game SMEs tend to turn into low-credit companies and their performance at $k = 0.02$ is relatively slow. Under the government's relatively high penalty cost $k = 0.03$, online game SMEs tend to turn into high-credit companies. This simulation result means that government regulations have a certain positive incentive effect on the behavioral evolution of online game SMEs.

Fig 7-2 shows the simulation behavior of credit providers under different government penalty cost constraints. In the same way, the credit provider also shows a preference for different penalty costs. In the case of $k = 0.01$ and $k = 0.02$, the credit provider's bias has changed to a non-professional credit provider, and the performance at $k = 0.02$ is relatively slow. Under the government's relatively high penalty cost $k = 0.03$, the credit provider tends to transform into a professional credit provider. This simulation result means that government regulation also has a certain positive incentive effect on the credit provider's behavior evolution.

Fig 8 shows the evolution simulation results of online game SMEs and credit providers under low government penalty costs and high government penalty costs. The main parameters $\omega = 0.5$, $U = 1.5$, $e_1 = 0.3$, $e_2 = 0.3$, $c_1 = 0.2$, $c_2 = 0.3$, and $x$ represent online game SMEs, $y$ represents the credit provider.

Fig 8-1 shows the simulation results under low government penalty costs ($k = 0.01$). Credit providers and online game SMEs tend to turn into low credit capacity and non-professional credit providers under this low government penalty cost, and credit providers are more active.

Fig 8-2 shows the simulation results under high government penalty costs ($k = 0.02$). Online game SMEs and credit providers tend to transform into high credit capability and professional credit providers under the relatively high government penalty costs. And online game SMEs have shown greater enthusiasm for becoming a demander of high-credit financing.

Fig 8-3 shows the simulation results under a higher level of government penalty costs ($k = 0.03$). Online game SMEs and credit providers have shown great enthusiasm for becoming a high-credit ability and professional credit provider under this very high government penalty cost. And the difference in the enthusiasm of the two to become high-quality credit providers is reduced.

## Conclusions and recommendations

### Research results

In order to promote the sustainable operation of online game SMEs, a comparative study of their intangible assets shows that the NEEQ delisting and ST online game SMEs still have some valuable intangible assets. If the credit provider and the online game SMEs establish a credit synergy mechanism through the intangible assets of the game companies, the financing problem of the online game SMEs can be solved to a certain extent. This paper constructs a dynamic game matrix between online game SMEs and their external credit providers. Online game SMEs and credit providers can choose to become high-credit and low-credit companies,

professional and non-professional credit providers, and credit synergy and benefit sharing coefficient affect the behavioral choices of online game companies and credit providers. Higher credit synergies and reasonable income sharing coefficients promote the transformation of online game SMEs and credit providers with high credit capability companies and professional credit providers. With unreasonable credit sharing benefits, credit providers are more inclined to transform into non-professional credit entities than online game SMEs. With reasonable credit sharing benefits, online game SMEs are more inclined to transform into parties with high credit capabilities than credit providers.

This conclusion shows that the credit provider and the financing demander tend to seek advantages and avoid disadvantages. When the credit environment is poor, the credit provider takes the lead and transforms into a non-professional credit provider. When the credit environment is good, online game SMEs are one step ahead of being a party with high credit ability to obtain credit supply from a credit provider and promote the transformation of a credit provider into a professional credit party. Similar to the case of reasonable credit synergies, when the income distribution coefficient is reasonable, online game SMEs also lead the credit provider to transform into a party with high credit ability. However, there are two possibilities in the case of an unreasonable credit distribution coefficient: With a low unreasonable credit allocation coefficient, online game SMEs' leading credit providers have turned into a low-credit capacity party; Second, with a high unreasonable credit allocation coefficient, credit suppliers have turned into non-professionals credit provider. This shows that the extremes of the two credit allocations make online game SMEs and credit providers show greater differences in the conversion of low-credit ability companies and non-professionals.

Different government punitive costs are based on the government's regulatory policies on relevant specific enterprises and entities related to credit relations. Under different government punitive cost level, online game SMEs have the same performance as credit providers, that is, the low government punitive costs show a tendency to transform to low credit capabilities and non-professional credit entities, and the high government punitive costs have both a tendency to transform into a subject with high credit ability and professional credit. With low government penalty costs, credit providers are more inclined to turn into non-professional than online game SMEs. Under relatively high government penalty costs, credit providers have a slower time as a professional credit subject than online game SMEs and even appear to be cautious and stressful. With very high government penalties, credit providers and online game SMEs are actively transforming into high-credit capacity companies and professional credit providers. The entire evolutionary game process shows that the credit providers dominate the credit selection, the credit provider shows more tendency to seek advantages and avoid disadvantages than online game SMEs.

## Administration recommendations

When government policies regulate the behavior of online game SMEs and credit suppliers and promote the development of the capital market, they can build a credit synergy mechanism to promote that online game SMEs and credit suppliers to be highly credit-worthy and professional credit providers. For example, in the online game industry, a credit synergy mechanism is built through large online game companies and SMEs, mergers and acquisitions institutions and SMEs, and banks and SMEs.

In addition, combined with relevant government regulations, such as the government's punitive measures for banks, large online game companies, mergers and acquisitions institutions, third-party financial institutions and other credit providers and online game SMEs, promote it to become a highly credit-worthy and professional credit provider. The dynamic

simulation evolution game vividly describes the behavioral choices of online game SMEs and credit providers and the motivation behind them, as well as the necessity of establishing a credit coordination mechanism and implementing government regulations.

## Acknowledgments

We are grateful to thank several professors from the School of Business Administration and some entrepreneurs for helping to discuss the business interaction, as well as our alumni and friends who helped with data collection. We also thank the editor and series editor for constructive criticisms of an earlier version.

## Author Contributions

**Data curation:** Gaige Tu.

**Formal analysis:** Gaige Tu.

**Investigation:** Gaige Tu.

**Methodology:** Gaige Tu.

**Project administration:** Gaige Tu.

**Resources:** Gaige Tu.

**Software:** Gaige Tu.

**Supervision:** Gaige Tu.

**Validation:** Gaige Tu.

**Visualization:** Gaige Tu.

**Writing – original draft:** Gaige Tu.

**Writing – review & editing:** Hao Chen, Chunxiao Zhu.

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
