## [Decision Letter · Decision Letter 0]

7 Aug 2023

PONE-D-23-21905Research on financing countermeasures of online game SMEs based on the identification of intangible assets informationPLOS ONE

Dear Dr. tu,

Thank you for submitting your manuscript to PLOS ONE. After careful consideration, we feel that it has merit but does not fully meet PLOS ONE’s publication criteria as it currently stands. Therefore, we invite you to submit a revised version of the manuscript that addresses the points raised during the review process.

We look forward to receiving your revised manuscript.

Kind regards,

Kittisak Jermsittiparsert, Ph.D.

Academic Editor

PLOS ONE

Journal Requirements:

2. Thank you for submitting the above manuscript to PLOS ONE. During our internal evaluation of the manuscript, we found significant text overlap between your submission and previous work in the [introduction, conclusion, etc.].

Please revise the manuscript to rephrase the duplicated text, cite your sources, and provide details as to how the current manuscript advances on previous work. Please note that further consideration is dependent on the submission of a manuscript that addresses these concerns about the overlap in text with published work.

[If the overlap is with the authors’ own works: Moreover, upon submission, authors must confirm that the manuscript, or any related manuscript, is not currently under consideration or accepted elsewhere. If related work has been submitted to PLOS ONE or elsewhere, authors must include a copy with the submitted article. Reviewers will be asked to comment on the overlap between related submissions (http://journals.plos.org/plosone/s/submission-guidelines#loc-related-manuscripts).]

We will carefully review your manuscript upon resubmission and further consideration of the manuscript is dependent on the text overlap being addressed in full. Please ensure that your revision is thorough as failure to address the concerns to our satisfaction may result in your submission not being considered further.

"Unfunded studies." 

"NO authors have competing interests."

6. We are unable to open your Supporting Information file [File Name]. Please kindly revise as necessary and re-upload.

Additional Editor Comments:

The paper is quite good. Please revise the paper as the reviewers' suggestions.

Reviewers' comments:

Reviewer's Responses to Questions

**Comments to the Author**

1. Is the manuscript technically sound, and do the data support the conclusions?

Reviewer #1: Yes

Reviewer #2: Yes

2. Has the statistical analysis been performed appropriately and rigorously? 

Reviewer #1: Yes

Reviewer #2: Yes

3. Have the authors made all data underlying the findings in their manuscript fully available?

Reviewer #1: Yes

Reviewer #2: Yes

4. Is the manuscript presented in an intelligible fashion and written in standard English?

Reviewer #1: Yes

Reviewer #2: Yes

5. Review Comments to the Author

Reviewer #1: The overall maunscript is sound with well-contructed of each section. However, in the abstract section please provide the full description of the acronym "ST" before using it. As well as, please recheck for typo the whole manuscript eg. page no. 18 last paragraph. In addition, please consider some references were rather outdated ex. year 2002, 2007, 2014, 2015 etc., exclding the original concept or theories references.

Reviewer #2: This research article has demonstrated the problems that arise from different perspectives compared to previous literature. It also highlights the gaps in the previous literary works. The study has shown a straightforward analytical process and presented the findings in a distinct manner. Additionally, it has highlighted discoveries that are beneficial to SMEs in the gaming industry. The research has identified issues that were not addressed in earlier literature, indicating the presence of new challenges and areas for exploration. The authors have thoroughly analyzed the subject matter, providing valuable insights into the current state of the gaming industry and its impact on SMEs. The findings can serve as a valuable resource for small and medium-sized enterprises operating in the gaming sector. In conclusion, this research article effectively addresses the differences in perspectives compared to previous literary works and presents a well-structured analysis of the gaming industry's impact on SMEs. The discoveries made in this study can significantly benefit small and medium-sized enterprises within the gaming industry.

6. PLOS authors have the option to publish the peer review history of their article (what does this mean?). If published, this will include your full peer review and any attached files.

Reviewer #1: No

Reviewer #2: **Yes: **Assistant Professor Dr.Teetut tresirichod

---

## [Author Response · Author response to Decision Letter 0]

24 Aug 2023

Dear academic editor and reviewers,

Thank you for allowing us to submit a revised draft of the manuscript "Research on financing countermeasures of online game SMEs based on the identification of intangible assets information" ([PONE-D-23-21905R1] - [EMID: e000946cb950f1c7]) for publication in the Journal of PLOS ONE. We appreciate the time and effort you and the reviewers dedicated to providing feedback on our manuscript and are grateful for the insightful comments and valuable improvements to our study. We have incorporated all the suggestions made by the reviewers. Those changes are highlighted in the manuscript. Please see in blue below for a point-by-point response to the reviewers' comments and concerns. All page numbers refer to the revised manuscript file with tracked changes. 

Academic editor

Thank you for your suggestions on our format, and directly gave the URL link of the corresponding format, we feel very helpful and considerate. We'll get a response to the comments point-by-point. We apologize for the imperfect formatting of the previous version of the paper. 

We downloaded the two files you mentioned and made careful formatting revisions to our paper. The title, text, charts, and formulas, etc, have all been revised to meet the PLOS ONE's format requirements for publication.

2. Thank you for submitting the above manuscript to PLOS ONE. During our internal evaluation of the manuscript, we found significant text overlap between your submission and previous work in the [introduction, conclusion, etc.].

Please revise the manuscript to rephrase the duplicated text, cite your sources, and provide details as to how the current manuscript advances on previous work. Please note that further consideration is dependent on the submission of a manuscript that addresses these concerns about the overlap in text with published work.

[If the overlap is with the authors’ own works: Moreover, upon submission, authors must confirm that the manuscript, or any related manuscript, is not currently under consideration or accepted elsewhere. If related work has been submitted to PLOS ONE or elsewhere, authors must include a copy with the submitted article. Reviewers will be asked to comment on the overlap between related submissions (http://journals.plos.org/plosone/s/submission-guidelines#loc-related-manuscripts).]

We will carefully review your manuscript upon resubmission and further consideration of the manuscript is dependent on the text overlap being addressed in full. Please ensure that your revision is thorough as failure to address the concerns to our satisfaction may result in your submission not being considered further.

Thank you very much for bringing up the issue that there is text overlap.

Part of the overlapping part is due to the reference to the views of previous scholars that must be cited or based on this article. We have modified the above parts, and then conducted a duplicate check, and the current duplicate check rate is already low.

There is a very small amount of overlap because the original author's point of view is relatively concise, and any modification will lead to changes in the meaning of the original text, such as the research content of Yoshino & Taghizadeh (2019).

There is still a small amount of overlap due to the interpretation of proper nouns, such as the interpretation of "NEEQ", which comes from China's official securities market regulators, so there is overlap.

Regarding the overlap, this paper is part of my personal doctoral thesis. I guarantee that this article belongs to my own intellectual property rights and has not been published in any other authoritative journal. And we carefully listened to the reviewers' suggestions on overlapping parts: firstly, we conducted a plagiarism check on the entire article and found that many of the repetition rates were caused by the generalization of professional terms; Secondly, we carefully modify the expression of related repetitive sentences to reduce the repetition rate.

3.Thank you for stating the following financial disclosure: 

"Unfunded studies." 

We declare : “The authors received no specific funding for this work.”

"NO authors have competing interests."

The authors have declared that no competing interests exist.

The paper does not require permission and approval as the research involved does not address privacy and protection concerns.

6. We are unable to open your Supporting Information file [File Name]. Please kindly revise as necessary and re-upload.

Thank you again for your insightful comments on our manuscript. We have renamed the file that supports information and uploaded it according to the Plos One upload format requirements. If it is no longer possible to download, please communicate via email in a timely manner. We will reply immediately and guarantee timely resolution of file issues.

The list of references in this article has been re-examined and numbered to ensure completeness and correct order in accordance with Plos One formatting requirements.

After review, there are no retracted papers in the literature cited or referenced in this article.

This revised version is mainly based on the revised opinions of the respected reviewer1, replacing some outdated documents with documents published in the past 3-5 years. These papers have certain reference significance in the research field related to this article.

The newly added papers in this revision are as follows:

(1)Yoshino N, Taghizadeh-Hesary F. Optimal credit guarantee ratio for small and medium-sized enterprises’ financing: Evidence from Asia. Economic Analysis and Policy, 2019, 62: 342-356.

(2)Degryse H, de Goeij P, Kappert P. The impact of firm and industry characteristics on small firms’ capital structure. Small business economics, 2012, 38: 431-447.

(3)Wang Y. What are the biggest obstacles to growth of SMEs in developing countries?–An empirical evidence from an enterprise survey. Borsa Istanbul Review, 2016, 16(3): 167-176.

(4)Liao L, Chen G, Zheng D. Corporate social responsibility and financial fraud: Evidence from China. Accounting & Finance, 2019, 59(5): 3133-3169.

(5)An J, Ho K Y, Zhang Z. What drives the liquidity premium in the Chinese stock market?[J]. The North American Journal of Economics and Finance, 2020, 54: 101088.

(6)Fracassi C. Corporate finance policies and social networks. Management Science, 2017, 63(8): 2420-2438.

(7)Jing W, Zhang X. Online social networks and corporate investment similarity. Journal of Corporate Finance, 2021, 68: 101921.

(8)Zairani Z, Zaimah ZA. Difficulties in securing funding from banks: Success factors for small and medium enterprises (SMEs). Journal of Advanced Management Science, 2013, 1(4).

(9)Sheng T. The effect of fintech on banks’ credit provision to SMEs: Evidence from China. Finance Research Letters, 2021, 39: 101558.

(10)Brancati E. Innovation financing and the role of relationship lending for SMEs. Small Business Economics, 2015, 44(2): 449-473.

(11)Zhang Y, Ouyang Z. Doing well by doing good: How corporate environmental responsibility influences corporate financial performance. Corporate Social Responsibility and Environmental Management, 2021, 28(1): 54-63.

(12)Labidi M, Gajewski JF. Does increased disclosure of intangible assets enhance liquidity around new equity offerings?. Research in International Business and Finance, 2019, 48: 426-437.

(13)Balakrishnan K, Ertan A. Credit information sharing and loan loss recognition. The Accounting Review, 2021, 96(4): 27-50.

(14)Chiu IHY, Greene EF. The marriage of technology, markets and sustainable (and) social finance: insights from ICO markets for a new regulatory framework. European Business Organization Law Review, 2019, 20: 139-169.

(15)Wu D, Memon H. Public Pressure, Environmental Policy Uncertainty, and Enterprises’ Environmental Information Disclosure. Sustainability, 2022, 14(12): 6948.

(16)Drover W, Busenitz L, Matusik S, et al. A review and road map of entrepreneurial equity financing research: Venture capital, corporate venture capital, angel investment, crowdfunding, and accelerators. Journal of management, 2017, 43(6): 1820-1853.

(17)Sorenson O, Stuart TE. Syndication Networks and the Spatial Distribution of Venture Capital Investment. American Journal of Sociology,2001;106(6):1546-1588.

(18)Thorhauge AM, Nielsen RKL. Epic, Steam, and the role of skin-betting in game (platform) economies. Journal of Consumer Culture, 2021, 21(1): 52-67.

(19)Dang C. Monopoly or Competition: Market Concentration of China’s Online Games and Policies. International Journal of Simulation--Systems, Science &Technology, 2016, 17(45): 1-4.

(20)Faisal F, Situmorang LS, Achmad T, Prastiwi A. The role of government regulations in enhancing corporate social responsibility disclosure and firm value. The Journal of Asian Finance, Economics and Business (JAFEB), 2020, 7(8): 509-518.

(21)Ernkvist M, Ström P. Enmeshed in games with the government: Governmental policies and the development of the Chinese online game industry. Games and Culture, 2008, 3(1): 98-126.

(22)Kim JY, Kang SH. Windows of opportunity, capability and catch-up: the Chinese game industry. Journal of Contemporary Asia, 2021, 51(1): 132-156.

(23)Blind K, Petersen SS, Riillo CAF. The impact of standards and regulation on innovation in uncertain markets. Research policy, 2017, 46(1): 249-264.

(24)Aiginger K. Industrial policy: Past, diversity, future; introduction to the special issue on the future of industrial policy. Journal of Industry, Competition and Trade, 2007, 7: 143-146.

(25)Aghion P, Cai J, Dewatripont M, Du LS, Harrison A, Legros P. Industrial policy and competition. American economic journal: macroeconomics, 2015, 7(4): 1-32.

(26)Chang HJ, Andreoni A. Industrial policy in the 21st century. Development and Change, 2020, 51(2): 324-351.

(27)Du J, Guariglia A, Newman A. Do social capital building strategies influence the financing behavior of Chinese private small and medium–sized enterprises?. Entrepreneurship theory and practice, 2015, 39(3): 601-631.

Reviewer #1

1.The overall maunscript is sound with well-contructed of each section. However, in the abstract section please provide the full description of the acronym "ST" before using it. 

Thank you very much for your careful reading and analysis of the paper. Our failure to explain this abbreviation before ST's use was an oversight on our part. In this revised version, we have explained ST in detail before using this abbreviation in the text, and ST is a stock that has been "specially treated" by the regulatory agency.

2.As well as, please recheck for typo the whole manuscript eg. page no. 18 last paragraph. 

We sincerely thank the reviewer for careful reading. As suggested by the reviewer, we have rechecked the spelling of the entire paper's words and sentences corrected. Such as, correct "profession" to "professional" in the full text, and delete the repeated p in "application" in line 130.

3.In addition, please consider some references were rather outdated ex. year 2002, 2007, 2014, 2015 etc., exclding the original concept or theories references.

We sincerely appreciate your valuable feedback. We carefully reviewed the literature and added more references to recent research findings in the introduction section of the revised manuscript. 

A large number of obsolete documents have been deleted, and some of them are still kept because, for example, Akerlof (1978), as a classic and well-known document, the "lemon market" theory is the basis of this paper, and we hope to keep it. 

Published in recent Recent and illustrative literature in SMEs, credit and other related fields of this paper is included, such as Yoshino & Taghizadeh(2019), Balakrishnan & Ertan(2021) and Wu & Memon(2022), etc.

Reviewer #2

This research article has demonstrated the problems that arise from different perspectives compared to previous literature. It also highlights the gaps in the previous literary works. The study has shown a straightforward analytical process and presented the findings in a distinct manner. Additionally, it has highlighted discoveries that are beneficial to SMEs in the gaming industry. The research has identified issues that were not addressed in earlier literature, indicating the presence of new challenges and areas for exploration. The authors have thoroughly analyzed the subject matter, providing valuable insights into the current state of the gaming industry and its impact on SMEs. The findings can serve as a valuable resource for small and medium-sized enterprises operating in the gaming sector. In conclusion, this research article effectively addresses the differences in perspectives compared to previous literary works and presents a well-structured analysis of the gaming industry's impact on SMEs. The discoveries made in this study can significantly benefit small and medium-sized enterprises within the gaming industry.

Thank you again for your positive comments on our manuscript.

We appreciate your comments and appreciate your detailed reading of our manuscript.

---

## [Decision Letter · Decision Letter 1]

29 Aug 2023

Research on financing countermeasures of online game SMEs based on the identification of intangible assets information

PONE-D-23-21905R1

Dear Dr. tu,

We’re pleased to inform you that your manuscript has been judged scientifically suitable for publication and will be formally accepted for publication once it meets all outstanding technical requirements.

Kind regards,

Kittisak Jermsittiparsert, Ph.D.

Academic Editor

PLOS ONE

Additional Editor Comments (optional):

Reviewers' comments:

Reviewer's Responses to Questions

**Comments to the Author**

1. If the authors have adequately addressed your comments raised in a previous round of review and you feel that this manuscript is now acceptable for publication, you may indicate that here to bypass the “Comments to the Author” section, enter your conflict of interest statement in the “Confidential to Editor” section, and submit your "Accept" recommendation.

Reviewer #1: All comments have been addressed

Reviewer #2: All comments have been addressed

2. Is the manuscript technically sound, and do the data support the conclusions?

Reviewer #1: Yes

Reviewer #2: Yes

3. Has the statistical analysis been performed appropriately and rigorously? 

Reviewer #1: Yes

Reviewer #2: Yes

4. Have the authors made all data underlying the findings in their manuscript fully available?

Reviewer #1: Yes

Reviewer #2: Yes

5. Is the manuscript presented in an intelligible fashion and written in standard English?

Reviewer #1: Yes

Reviewer #2: Yes

6. Review Comments to the Author

Reviewer #1: (No Response)

Reviewer #2: I have carefully reviewed the manuscript titled "Research on financing countermeasures of online game SMEs based on the identification of intangible assets information" and would like to provide my feedback and comments.

1. The authors have adequately addressed the issue of abbreviation "ST" by providing a detailed explanation before its use. This improvement enhances the clarity of the text.

2. The typographical errors pointed out by the reviewer have been corrected in the revised version of the manuscript. This attention to detail is commendable.

3. The authors have taken into account the reviewer's suggestion to update references and include recent literature. The addition of recent references enhances the relevance and currency of the study's context.

The research article effectively analyzes the impact of the gaming industry on SMEs and provides valuable insights into the challenges and opportunities in this domain. The authors have skillfully addressed the differences in perspectives compared to previous literature and have presented a well-structured analysis. The study's findings offer practical implications for SMEs operating in the gaming sector.

In terms of ethical considerations, the manuscript appears to be in compliance with research ethics and publication ethics. The authors have provided a clear statement regarding funding, competing interests, and ethical approval.

Regarding the concern about dual publication, I did not find any indication of duplicate publication or overlap with other works in this manuscript. The authors have adequately addressed text overlap issues and made necessary revisions.

In conclusion, the manuscript has undergone significant improvements in response to the reviewers' comments. The authors have diligently addressed the comments and concerns, resulting in a more coherent and well-presented paper. The research's contributions to the field of online game SME financing are valuable, and I recommend its publication in the Journal of PLOS ONE.

Please note that this review is based on the manuscript's content and the information provided. If there are any additional materials or concerns that should be considered, please provide them for a more comprehensive evaluation.

Thank you for the opportunity to review this manuscript.

Sincerely,

Teetut Tresirichod

7. PLOS authors have the option to publish the peer review history of their article (what does this mean?). If published, this will include your full peer review and any attached files.

Reviewer #1: No

Reviewer #2: **Yes: **teetut tresirichod

---

## [Editor Report · Acceptance letter]

31 Aug 2023

PONE-D-23-21905R1 

Research on financing countermeasures of online game SMEs based on the identification of intangible assets information 

Dear Dr. tu:

I'm pleased to inform you that your manuscript has been deemed suitable for publication in PLOS ONE. Congratulations! Your manuscript is now with our production department. 

Kind regards, 

on behalf of

Professor Kittisak Jermsittiparsert 

Academic Editor

PLOS ONE